# Disposal practices of unused and expired pharmaceuticals among the general public in Quetta city, Pakistan

**Muhammad Shoaib**[1], **Abdul Raziq**[2], **Qaiser Iqbal**[1], **Fahad Saleem**[1], **Sajjad Haider**[1], **Rabia Ishaq**[1], **Zaffar Iqbal**[3], **Mohammad Bashaar**[4] *

1 Faculty of Pharmacy & Health Sciences, University of Baluchistan, Quetta, Pakistan, 2 Department of Statistics, University of Baluchistan, Quetta, Pakistan, 3 Health Department, Government of Baluchistan, Quetta, Pakistan, 4 SMART Afghan International Trainings & Consultancy, Kabul, Afghanistan

☯ These authors contributed equally to this work.
* dr.mbashaar@gmail.com

**Data Availability Statement:** Data cannot be shared publicly because of restrictions by Advance Studies & Research Board, University of Balochistan, Quetta. Data are available from the Chairman, Advance Studies & Research Board

## Abstract

### Objective

Although community practices towards disposal of expired and unused medications vary globally, the phenomenon is neglected specifically in developing countries. We, therefore, aimed to assess the disposal practices of unused and expired pharmaceuticals among the general public in Quetta city, Pakistan.

### Methods

A questionnaire-based, cross-sectional survey was conducted among 830 respondents in Quetta city, Pakistan. A pre-validated, structured questionnaire was used for data collection. Data were coded and analyzed by Statistical Package for Social Science (SPSS) version 26. Both descriptive and inferential statistics were sued and p<0.05 was taken as significant.

### Results

Almost 87% of the respondents had unused medicines at their homes and reusing the medications was the purpose of medicine storage (50%). Medicines were mostly kept in refrigerators (36.0%) followed by bedrooms (28.8%). Fifty percent of the respondents never checked the expiry date before using the medications. The procedure to dispose of nearly expired or expired medicines was unknown to 88% of the respondents and for those reasons, medicines were disposed either in household trash or flushed in the toilet or sink. Interestingly, medicines were kept at home even after being expired by 27% of the respondents. Income was associated with reason of storing medicines (p = 0.004; φc = 0.402) while educational status had significant associations with storage of medications (p = 0.002; φc = 0.399), reading about storage instructions (p = 0.008; φc = 0.452) and checking expiry date before purchasing and using the medicines (p = 0.007; φc = 0.114 and p = 0.001; φc = 0.100) respectively.

(gso@um.uob.edu.pk) for researchers who meet the criteria for access to confidential data.

**Funding:** authors received no specific funding for this work.

**Competing interests:** The authors have declared that no competing interests exist.

## Conclusion

Improper storage and disposal of unused and expired medications is a common practice among study respondents and raised serious concerns. Findings of the current study call upon immediate development and implementation of the strategies to promote awareness and strengthen the pharmaceutical waste management program at the societal level.

## Background

In recent decades, the increased prevalence of both acute and chronic medical conditions has increased medicine usage. Even though pharmaceutical use is on the rise, it is reported that patients do not consume their medications and such unused medicines are stored in households [1, 2]. Additionally, in clinical practice, 50% of the medications are prescribed irrationally and that results in the redundant storage of medicines causing further harm to the patients as well as the environment [3]. The World Health Organization also revealed that as patients do not use their medicines regularly, their related friends and families are often in contact with medications and can utilize those in time of need without a professional consultation [4]. Moreover, with the worry of unused pharmaceuticals, expired medicines in the possessions of the public are another grave concern. Unfortunately, the safe disposal of such medicines is poorly practiced in societies [4].

The existence of unused or expired pharmaceuticals in homes always places an awaiting hazard. The intentional or unintentional use of expired medicines is not only lethal for humans but is also dangerous to the atmosphere [5–7]. Within this context, expired medicines are often trashed in normal bins or are discarded in sewerage and this incidence when added to the spillways or drinking water places a serious threat to the normal fauna and flora [3]. A classic example of such obliteration is the development of renal failure among vultures. On investigation, it was revealed that carcasses were contaminated with Diclofenac Sodium resulting in the development of kidney-related issues among scavengers [8]. Besides, the traces of Ethinyl Estradiol in water resulted in blight sexual growth and the feminization of marine life [9]. Other than pathological issues, Bashaar et al reported that offshoots after a vaccination movement when disposed of in a local facility caused contagious and physical injuries to the people looking for useful things in the trash [3]. Therefore, in recent years, this issue has caught the serious attention of the policymakers, civil activists, and other health and environmental agencies [10].

In terms of medicine management, it was reported that many of the healthcare facilities in developing countries either lack ignition capability or do not have a waste management system [11, 12]. Keeping the situation in view, WHO came up with an agenda to develop standard operating procedures (SOPs) to address the issues of waste management in the developing world [13]. However, even today, most of the countries do not have SOPs for the clearance of unwanted and fallow pharmaceuticals. Specifically talking about Pakistan, the Drug Regulatory Authority (DRAP) is held conscientious for the proficient scrutinization and assessment of medicine use in the country. Notwithstanding, these policies are weakly implemented and have deficits, and in general the medicine dumping is considered dysfunctional. Possible reasons for this unfortunate management are deficient resources, corruption, and poor accomplishment of governmental rules & regulations. Therefore, in addition to policy implementation, understanding this issue by the public is equally important. Based on our review, we were unable to find any study that focused on expired medicine management in Balochistan, and no information is accessible about the clearance information of run-out

pharmaceuticals in Quetta city. Hence, in the light of the above discussion, it is evident that the practices of the general population towards medicine disposal should be addressed. Therefore, the current study is aimed to assess the dumping of unused and run-out medicines among the common population in Quetta city, Pakistan.

## Methods

### Study design and settings

This was a questionnaire-based, cross-sectional assessment. Data were collected from patients attending the general outpatient departments (OPDs) of Sandeman Provincial Hospital, Quetta (SPHQ). Located in the center of the city, SPHQ was established in 1939. It is a tertiary public care, teaching, and government hospital and is an institute of choice for most of the population [14].

### Study population, criteria, and sampling

Eligible participants were adults aged 18 or over, and of either gender visiting SPHQ for treatment or consultation were approached. Consenting to participate and an understanding of Urdu (lingua franca of Pakistan) was needed by the respondents. However, immigrants from other countries and with mental impairments were excluded. Proportionate-based sampling with a double design effect method was employed for sample size calculation and 830 respondents were needed that were approached conveniently for data collection [15].

n = $Z^2$ x p (1-p) / $d^2$ x DEFF; *(n = sample size, Z = confidence interval, p = expected proportion, d = margin of error and DEFF = design effect)*

n = $1.96^2$ x 0.50 (1–0.50) / $0.05^2$ x 2

n = 754 x 0.10 (drop out) = 830

### Study instrument and validation

We did a thorough review of the literature and developed the initial version of the questionnaire in the English language [3, 6, 16–18]. In addition to the demographic information, the questionnaire assessed knowledge and practices towards unused and expired medicines. The forward-backward method was used to translate the questionnaire into Urdu by bilingual experts [19]. The internal consistency ($\alpha = 0.850$) was established through Cronbach's alpha test [20]. For face and content validity, the questionnaire was reviewed by experts and pretested on 15 respondents [21]. Minor changes were highlighted during the pilot phase and the questionnaire was made available for the principal study (S1 File). Data from the pilot phase was not utilized afterward.

### Data collection and analysis

Data were collected from patients attending the outpatient department (OPD) of the Sandeman Provincial Hospital (SPH) Quetta, Pakistan. Data were coded and added to SPSS v 26.0 for analysis. Based on the objectives of the study, descriptive analysis was conducted initially. Relationships were observed by Chi-Square test and Phi / Cremer were used for interpretation where applicable. For all analyses, $p<0.05$ was considered significant.

### Ethical approval

The Advanced Study and Research Board, University of Balochistan, Quetta approved the study. Permission was also taken from the Medical Superintendent of SPHQ. Written

informed consent was obtained from all the respondents. Participation in this research was voluntary and the identity was kept confidential.

## Results

### Demographic profile of the study respondents

Eight hundred and thirty responded to the study with a mean age of 31.31 ± 10.25 years. The gender distribution was almost equal (50%), while 46.6% had a higher secondary level of education. Eighty-three percent of the medications were purchased over the counter while 65.3% were purchased through prescription (Table 1).

### Storage, and disposal practices of unused medicines

Almost 87% of the respondents had unused medicines at their homes and reusing the medications was the purpose of medicine storage (50%). Most of the respondents read storage instructions on the labels and were aware of the storage methods. Medicines were mostly kept in refrigerators (36.0%) followed by bedrooms (28.8%). Only 55 respondents disposed of

**Table 1. Demographic profile of the study respondents.**

| Characteristics | Frequency (N) | Percentage (%) |
|---|---|---|
| *Age (31.31±10.25)* | | |
| 18–27 | 370 | 44.6 |
| 28–37 | 223 | 26.9 |
| 38–47 | 150 | 18.1 |
| >47 | 87 | 10.5 |
| *Gender* | | |
| Male | 409 | 49.3 |
| Female | 421 | 50.7 |
| *Marital status* | | |
| Married | 417 | 50.2 |
| Unmarried | 413 | 49.8 |
| *Locality* | | |
| Rural | 422 | 50.8 |
| Urban | 408 | 49.2 |
| *Education* | | |
| Secondary | 338 | 40.7 |
| Higher secondary | 387 | 46.6 |
| Graduate | 105 | 12.6 |
| *Occupation* | | |
| Unemployed | 251 | 30.2 |
| Housewife | 311 | 37.4 |
| Public employee | 199 | 23.9 |
| Business | 69 | 8.3 |
| *Income** | | |
| None | 562 | 67.7 |
| > 20000 | 92 | 11.0 |
| 20001–40000 | 95 | 11.4 |
| > 40000 | 81 | 9.7 |

*Pakistan Rupees

unused medications at their homes and trash cans (dustbins) were used by most respondents for disposal of unused medicines (Table 2).

## Practices and disposal of expired medicines

The practices and disposal of expired medicines are presented in Table 3. Most of the respondents (94.9%) checked the expiry date before procuring the medicines. However, nearly 50% never checked the expiry date before using the medications. The procedure to dispose of nearly expired or expired medicines was unknown to 88% of the respondents and for those reasons, medicines were disposed of either in household trash or flushed in the toilet or sink. Intriguingly, medicines were kept at home even after being expired by 27% of the respondents.

## Association between storage, and disposal practices of unused medicines and demographics

A significant association with moderate association ($p<0.05$; $\varphi c = 0.402$) was reported when the reasons for storing medicines were compared with income. Accordingly, education was also reported to have significant associations with the storage of medications and reading

**Table 2. Storage, and disposal practices of unused medicines.**

| Items | Frequency | Percentage |
|---|---|---|
| *Do you currently have any unused medicines stored at home?* | | |
| Yes | 719 | 86.6 |
| No | 111 | 13.4 |
| *Why do you keep unused medicines at home?* | | |
| Can reuse the medicine | 415 | 50.0 |
| Medicine is needed in an emergency | 356 | 42.8 |
| Can be used by friends or family members | 59 | 7.2 |
| *Do you know how to store medicines at home?* | | |
| Yes | 625 | 75.3 |
| No | 205 | 24.7 |
| *Do you read storage instructions on the labels/leaflets?* | | |
| Yes | 657 | 79.1 |
| No | 173 | 20.9 |
| *Where do you store your unused medicine?* | | |
| Kitchen cabinet | 178 | 21.4 |
| Bathroom cabinet | 85 | 10.2 |
| Bedroom cabinet | 245 | 28.8 |
| Medicine box | 23 | 2.77 |
| Refrigerator | 299 | 36.0 |
| *Do you dispose of unused medicines stored at your home?* | | |
| Yes | 55 | 6.7 |
| No | 775 | 93.3 |
| *How do you dispose of unused medicines stored at your home?* * | | |
| The exchange at the pharmacies | 10 | 18.1 |
| Throw away in dustbins (household trash) | 32 | 58.1 |
| Give to hospitals/clinic | 5 | 9.0 |
| Give to friends or relatives | 8 | 14.5 |

*Response out of 55

**Table 3. Practices and disposal of expired medicines.**

| Items | Frequency | Percentage |
|---|---|---|
| *Do you check the expiry date of the medicines before purchasing?* | | |
| Yes | 788 | 94.9 |
| No | 42 | 5.0 |
| *Do you check the expiry date before using the medicines?* | | |
| Yes | 405 | 48.7 |
| No | 425 | 51.3 |
| *Do you know the procedure to dispose of nearly expired medicines?* | | |
| Yes | 98 | 11.8 |
| No | 732 | 88.2 |
| *How do you dispose of nearly expired medicines?* | | |
| Donte to hospital/clinic | 85 | 10.2 |
| Return it to pharmacies | 61 | 7.3 |
| Throw away in dustbins (household trash) | 312 | 37.5 |
| Flush in toilet or sink | 301 | 36.2 |
| Give them to friends/family members/ others | 41 | 4.9 |
| Keep at home until expired | 30 | 3.6 |
| *Do you know the procedure to dispose of expired medicines?* | | |
| Yes | 98 | 11.8 |
| No | 732 | 88.2 |
| *How do you dispose expired medicines?* | | |
| Throw away in dustbins (household trash) | 354 | 42.6 |
| Flush in toilet or sink | 212 | 25.5 |
| Return them to pharmacies or hospitals for disposal | 37 | 4.4 |
| No action (keep at home) | 227 | 27.3 |

about storage instructions ($p < 0.05$). For both significant associations, the effect size was also moderate ($\varphi c = 0.399$ & $0.452$ respectively). No association was reported for other variables as shown in Table 4.

## Association between practices and disposal of expired medicines and demographics

The association between practices and disposal of expired medicines and demographics is reported in Table 5. As in Table 4, education was significantly associated with checking the expiry date of medicines before procurement as well as before using them. Weak effect size was reported for both variables ($\varphi c = 0.114$ & $0.100$ respectively). No association was reported for other variables Table 5.

## Discussion

Medicines are generally classified based on similarities. The most used criterion is the pharmacological class that further subdivides medicines based on the mode of action, the physiologic effects of the medicines and chemical structure [22]. According to the Food and Drug Administration (FDA), the best option of managing unused or expired medicines is to find a drug take-back location. Normally, the take-back location is a local pharmacy or a hospital. These facilities offer on-site medicine drop-off boxes, mail-back programs, or in-home disposal products. The other option in the absence of a take-back location is checking the FDA's Flush List. However, if the medicines are not in the Flush List, FDA recommends disposing the

**Table 4. Association between storage, and disposal practices of unused medicines and demographics.**

| Items | P-Value* | | | | | | |
|---|---|---|---|---|---|---|---|
| | Age | Gender | Status | Locality | Education | Occupation | Income |
| **Do you currently have any unused medicines stored at home?** | 0.254 | 0.554 | 0.411 | 0.091 | 0.177 | 0.488 | 0.288 |
| | φc = N/A | φc = N/A | φc = N/A | φc = N/A | φc = N/A | φc = N/A | φc = N/A |
| **Why do you keep unused medicines at home?** | 0.335 | 0.129 | 0.277 | 0.377 | 0.590 | 0.600 | **0.004** |
| | φc = N/A | φc = N/A | φc = N/A | φc = N/A | φc = N/A | φc = N/A | **φc = 0.402** |
| **Do you know how to store medicines at home?** | 0.399 | 0.445 | 0.519 | 0.148 | **0.002** | 0.272 | 0.516 |
| | φc = N/A | φc = N/A | φc = N/A | φc = N/A | φc = 0.399 | φc = N/A | φc = N/A |
| **Do you read storage instructions on the labels/leaflets?** | 0.088 | 0.200 | 0.551 | 0.199 | **0.008** | 0.126 | 0.500 |
| | φc = N/A | φc = N/A | φc = N/A | φc = N/A | φc = 0.452 | φc = N/A | φc = N/A |
| **Where do you store your unused medicine?** | 0.700 | 0.177 | 0.622 | 0.410 | 0.224 | 0.177 | 0.100 |
| | φc = N/A | φc = N/A | φc = N/A | φc = N/A | φc = N/A | φc = N/A | φc = N/A |
| **Do you dispose of unused medicines stored at your home?** | 0.118 | 0.514 | 0.418 | 0.841 | 0.281 | 0.547 | 0.109 |
| | φc = N/A | φc = N/A | φc = N/A | φc = N/A | φc = N/A | φc = N/A | φc = N/A |
| **How do you dispose of unused medicines stored at your home?** | 0.091 | 0.065 | 0.557 | 0.074 | 0.300 | 0.099 | 0.058 |
| | φc = N/A | φc = N/A | φc = N/A | φc = N/A | φc = N/A | φc = N/A | φc = N/A |

*Chi-Square test; Bold values represent significant associations

Note: φc = Phi/Cramer's V; N/A = Not applicable

medicine in trash before mixing them with unappealing substance such as dirt or cat litter. Medicines should not be crushed, and the mixture should be placed in a sealed plastic bag before disposing into the trash [23].

For decades, managing medicines (stored, unused, and expired) have received the attention of both healthcare professionals and social scientists. Additionally, with the increased frequency of diseases, medicines used around the globe has immensely increased. The IQVIA Institute for Human Data Science reported an increase of 3% compound annual growth rate in medicine use since 2014, and a projected increase of 2–5% of global spending on

**Table 5. Association between storage, and disposal practices of expired medicines and demographics.**

| Items | P-Value* | | | | | | |
|---|---|---|---|---|---|---|---|
| | Age | Gender | Status | Locality | Education | Occupation | Income |
| *Do you check the expiry date of the medicines before purchasing?* | 0.149 | 0.775 | 0.249 | 0.089 | **0.007** | 0.600 | 0.177 |
| | φc = N/A | φc = N/A | φc = N/A | φc = N/A | φc = 0.114 | φc = N/A | φc = N/A |
| *Do you check the expiry date before using the medicines?* | 0.559 | 0.250 | 0.320 | 0.706 | **0.001** | 0.735 | 0.553 |
| | φc = N/A | φc = N/A | φc = N/A | φc = N/A | φc = 0.100 | φc = N/A | φc = N/A |
| *Do you know the procedure to dispose of nearly expired medicines?* | 0.104 | 0.261 | 0.180 | 0.200 | 0.280 | 0.572 | 0.516 |
| | φc = N/A | φc = N/A | φc = N/A | φc = N/A | φc = N/A | φc = N/A | φc = N/A |
| *How do you dispose of nearly expired medicines?* | 0.338 | 0.419 | 0.447 | 0.109 | 0.119 | 0.422 | 0.408 |
| | φc = N/A | φc = N/A | φc = N/A | φc = N/A | φc = N/A | φc = N/A | φc = N/A |
| *Do you know the procedure to dispose of expired medicines?* | 0.574 | 0.077 | 0.122 | 0.240 | 0.310 | 0.470 | 0.283 |
| | φc = N/A | φc = N/A | φc = N/A | φc = N/A | φc = N/A | φc = N/A | φc = N/A |
| *How do you dispose of expired medicines?* | 0.630 | 0.931 | 0.149 | 0.801 | 0.411 | 0.099 | 0.142 |
| | φc = N/A | φc = N/A | φc = N/A | φc = N/A | φc = N/A | φc = N/A | φc = N/A |

*Chi-Square test; Bold values represent significant associations

Note: φc = Phi/Cramer's V; N/A = Not applicable

medications [24]. Although there is no specific data available from Pakistan on medicine expenditure, total health expenditure in 2017–18 reported an increase of 31.3% when compared with 2015–2016. It is pertinent to mention here that this increase is for direct health expenditures, and excludes health-related expenditures on training, research, and environmental health [25]. With such an enormous increase in healthcare system utilization, quality management of medicines always remains debatable [3]. Medicine management is a highly debatable topic around the globe [26–28], however, data from Pakistan is limited in literature and is reported from metropolitan areas of the country [18, 29, 30]. There is a paucity of information from remote areas of the country and for that reason, the current study was undertaken. To the best of our knowledge, this is the first study from Baluchistan (Quetta city), the largest of the provinces of Pakistan.

Possession of unused medicines was prevalent from the study findings that are in line with what is reported from studies of the same nature [3, 31]. On the contrary, where reusing the medicines was the reason for storage, literature has reported conflicting results [32, 33]. A possible explanation of this difference is related to the limitations faced by the healthcare system of Pakistan. Where inpatients are provided with ample facilities, outpatients must procure their medicines as public hospitals do not provide medicines to them. Medicine prices in the last years have increased considerably [34] and therefore most of the patients keep the unused medicines because of affordability issues. However, storing unused medicines does promote medicine exchange leading to self-medication which is a common practice in developing countries hence leading to absurd medicine use [31].

Refrigerators were used for medicine storage followed by bedroom cabinets (Table 2). As frequently reported in the literature [31, 33, 35, 36], both unused and expired medicines in the current study were also disposed either in dustbins or flushed into the sinks or toilets. Such practices are associated with ecological hazards that lead to a variety of concerns including antimicrobial resistance, damage to the aquatic environment, and injurious effects to the community health through drinking water [7]. Specifically in Quetta city where drain water is used for cultivation, this leads to contamination of vegetables and fruits which may contain medications in trace amounts [37, 38]. Even though the city has an advanced water treatment technology system, it is often unfunctional because of a lack of operational funding [39]. Disposing medicines in Quetta city is a challenge that needs priority attention to minimize the harmful effects caused by the addition of medicines in the environment. Even when the water is treated, most of the pharmaceutical compounds can enter water supply because conventional wastewater system fails to eliminate the pharmaceutical compounds completely [40]. Humans are directly affected by such compounds as oxidative stress is initiated through the consumption of untreated water. Various studies have revealed that oxidative stress is responsible for onset or progression of several diseases like atherosclerosis, metabolic disorders, cardiovascular diseases, diabetes, and cancer [41]. Free radicals are formed directly in response to oxidizing pollutants and results in oxidative stress that produces potential harm as discussed in literature [42].

Correlating our discussion, poor disposal practices in the current study are associated with a lack of information. The participants had no information on medicine disposal and that led to take uninformed decisions in disposing of their medications. For us, this is not surprising as pharmacies in Pakistan follow a business model and are not patient-oriented [43]. Furthermore, healthcare professionals (physicians and pharmacists) also pay the least attention to patient education and counseling. Lastly, little information is passed to the public by authorities (DRAP) on the disposal of unused or expired drugs that altogether complicates the issue. Our observations are supported by literature and the lack of information on medicine disposal methods is recorded from different countries [3, 31, 44].

While discussing the associations, education was found to be significantly related and the interpretation is relatively straightforward. Education plays a key role in developing the importance of medicines and treatment regimens, rational utilization of healthcare services, and an understanding of the needs and demands of pharmaceutical care [45]. Furthermore, education also maximizes the opportunity to exercise better control over the treatment regimens and disease conditions [46]. Within this context, Raghupathi & Raghupathi in their empirical assessment identified associations between education and health indicators. The authors targeted 26 OECD countries and data of 20 years (1995–2015) was analyzed [47]. It was concluded that people with higher educational achievement had better health and lifespans compared to less educated. Among all levels, tertiary education was critical and rated as an influencing factor. Therefore, the relationship of education is not surprising as educated people tend to read more about their diseased conditions, the medications they are using, and the methods while storing or dealing with unused or expired medicines.

Income was also significantly associated with the reason for storing medicines. We must remember that Pakistan faces a huge crisis when it comes to efficient and effective utilization of resources. Only 1.2% of the GDP is spent on health [48] and most of the expenses incurred during diagnosis, treatment, and management of diseases by the patients are out of pocket [49]. Besides, Pakistan's fragile economy was badly affected by COVID-19 and long-term effects increased unemployment because various sectors are still in crisis. Medicine prices in Pakistan have seen an increase of almost 300% in the last two years (highest in 40 years) and thus affordability of medicines for low-income earners is a grave concern [50]. With an increase in recession and poverty index, this seems to be a decent reason to store medicines to be reused. However, storing medicines has its limitations and this should be discouraged by the healthcare professionals while having interactions with their patients. Nevertheless, the conditions will remain the same until the Government of Pakistan starts investing in healthcare sector of Pakistan and show serious efforts in resolving the issues of medicine affordability in Pakistan.

An overwhelming finding of the current study revealed that 27% of the respondents kept expired medicines at their homes (Table 3). We must remember that storing expired medicines at homes may result in intentional or accidental consumption which can result in a health crisis [30]. Moreover, consuming expired medicines cause more serious illnesses because of change in chemical composition [51]. Keeping expired medicines at home is relatable to poor knowledge regarding safe disposal practices. Within this context, the guidelines proposed by internal agencies are intended for national authorities (DRAP in the case of Pakistan) and the least attention is given to household levels [52]. To circumvent a health crisis, there is an urgent need for a well-designed public education program focusing on medicine storage and disposal of expired medicines in Quetta city, Pakistan.

## Conclusion

Improper storage and disposal of unused and expired medications is a common practice among study respondents and raised serious concerns. Addressing this issue needs a coordinated and systematic public awareness campaigns initiated by DRAP to address the negative risks of unused and expired medicines. This will help in promoting reliable practices that will be beneficial for the society and healthcare system.

### Strengths and limitations

Storage and disposal practices of unused and expired medicines are reported for the first time from Quetta city, Pakistan. However, the study findings are not generalized and consequently

a large-scale study is recommended to highlight storage and disposal practices of unused and expired medicines at a large scale.

## Recommendations

We do agree that comprehensive information was retrievable from the respondents which was unexplored in the current study. To get an in-depth knowledge on unused and expired medicines, a home-based study is hereby recommended. This will exactly present the types and quantity of unused and expired medicines that will be in possession of the study respondents. This will also help in assessing the cost incurred and its effect on the demographics as well as on the society.

## Supporting information

**S1 File.**
(DOCX)

## Acknowledgments

We sincerely thank the participants for their contribution to the study. We also acknowledge the Medical Superintend of SPHQ for permission to conduct this study.

## Author Contributions

**Conceptualization:** Mohammad Bashaar.

**Data curation:** Muhammad Shoaib, Sajjad Haider.

**Formal analysis:** Abdul Raziq.

**Methodology:** Zaffar Iqbal.

**Resources:** Rabia Ishaq.

**Supervision:** Fahad Saleem, Mohammad Bashaar.

**Writing – original draft:** Qaiser Iqbal.

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
