## [Decision Letter · Decision Letter 0]

15 Feb 2022

PONE-D-22-00011Disposal practices of unused and expired pharmaceuticals among the general public in Quetta city, PakistanPLOS ONE

Dear Dr. Bashaar,

Thank you for submitting your manuscript to PLOS ONE. After careful consideration, we feel that it has merit but does not fully meet PLOS ONE’s publication criteria as it currently stands. Therefore, we invite you to submit a revised version of the manuscript that addresses the points raised during the review process.

We look forward to receiving your revised manuscript.

Kind regards,

Hongxun Tao

Academic Editor

PLOS ONE

Journal Requirements:

 [The funders had no role in study design, data collection and analysis, decision to publish, or preparation of the manuscript.] 

3. Please ensure that you include a title page within your main document. You should list all authors and all affiliations as per our author instructions and clearly indicate the corresponding author.

Reviewers' comments:

Reviewer's Responses to Questions

**Comments to the Author**

1. Is the manuscript technically sound, and do the data support the conclusions?

Reviewer #1: Yes

Reviewer #2: Yes

Reviewer #3: Yes

2. Has the statistical analysis been performed appropriately and rigorously? 

Reviewer #1: Yes

Reviewer #2: Yes

Reviewer #3: Yes

3. Have the authors made all data underlying the findings in their manuscript fully available?

Reviewer #1: Yes

Reviewer #2: Yes

Reviewer #3: Yes

4. Is the manuscript presented in an intelligible fashion and written in standard English?

Reviewer #1: Yes

Reviewer #2: Yes

Reviewer #3: Yes

5. Review Comments to the Author

Reviewer #1: The authors assessed the current disposal of unused and used up medicines by the general public in Quetta, Pakistan by means of a questionnaire, a typical but forgotten social health problem currently faced by the public.

1 Suggest some feasible measures for the management of expired drugs.

2 It is suggested to classify the types of drugs before investigating the disposal.

Reviewer #2: φc was reported in the paper. Whereas，originnal φc data was not supplied in the Tables. Could you provide them?

The average age of the responders was 31. No enough aging population was included. Dose the age distribution in the questionnaire survey idetified with population distribution of Pakistan?

More information could be collected, such as variety of the medicines unused and expired pharmaceuticals among the general public. These data should be valuable for policy setting.

Reviewer #3: The research is well designed, well written, and logically rigorous. However, the fit of the research topic with this journal needs to be improved. It is recommended that the author should provide necessary experimental evidence or literature support for the regulatory effects of compounds on oxidative stress, and strengthen the discussion on oxidative stress. It is recommended to accept the manuscript after major revision.

6. PLOS authors have the option to publish the peer review history of their article (what does this mean?). If published, this will include your full peer review and any attached files.

Reviewer #1: No

Reviewer #2: No

Reviewer #3: No

---

## [Author Response · Author response to Decision Letter 0]

3 Apr 2022

We are thankful to the editorial team of PLOS One for considering our manuscript. All queries of the reviewers are answered and a rebuttal is attached with the submission.

---

## [Decision Letter · Decision Letter 1]

25 Apr 2022

Disposal practices of unused and expired pharmaceuticals among the general public in Quetta city, Pakistan

PONE-D-22-00011R1

Dear Dr. Bashaar,

We’re pleased to inform you that your manuscript has been judged scientifically suitable for publication and will be formally accepted for publication once it meets all outstanding technical requirements.

Kind regards,

Hongxun Tao

Academic Editor

PLOS ONE

Additional Editor Comments (optional):

Reviewers' comments:

Reviewer's Responses to Questions

**Comments to the Author**

1. If the authors have adequately addressed your comments raised in a previous round of review and you feel that this manuscript is now acceptable for publication, you may indicate that here to bypass the “Comments to the Author” section, enter your conflict of interest statement in the “Confidential to Editor” section, and submit your "Accept" recommendation.

Reviewer #1: All comments have been addressed

Reviewer #3: All comments have been addressed

2. Is the manuscript technically sound, and do the data support the conclusions?

Reviewer #1: Yes

Reviewer #3: (No Response)

3. Has the statistical analysis been performed appropriately and rigorously? 

Reviewer #1: Yes

Reviewer #3: (No Response)

4. Have the authors made all data underlying the findings in their manuscript fully available?

Reviewer #1: Yes

Reviewer #3: (No Response)

5. Is the manuscript presented in an intelligible fashion and written in standard English?

Reviewer #1: Yes

Reviewer #3: (No Response)

6. Review Comments to the Author

Reviewer #1: The authors assessed the dumping of unused and run-out 128 medicines among the common population in Quetta city, Pakistan. The results are interesting.

Reviewer #3: NO COMMENT IN THIS VERSION, THE AUTHOR HAVE REVISED THE MANUSCRIPT WELL, I SUGGEST TO ACCEPT THIS PAPER IN ITS PRESENT FORM

7. PLOS authors have the option to publish the peer review history of their article (what does this mean?). If published, this will include your full peer review and any attached files.

Reviewer #1: No

Reviewer #3: No

---

## [Editor Report · Acceptance letter]

29 Apr 2022

PONE-D-22-00011R1 

Disposal practices of unused and expired pharmaceuticals among the general public in Quetta city, Pakistan 

Dear Dr. Bashaar:

I'm pleased to inform you that your manuscript has been deemed suitable for publication in PLOS ONE. Congratulations! Your manuscript is now with our production department. 

Kind regards, 

on behalf of

Dr. Hongxun Tao 

Academic Editor

PLOS ONE